# Correlative Light and Electron Microscopy (CLEM) Analysis of Nuclear Reorganization Induced by Clustered DNA Damage Upon Charged Particle Irradiation

**DOI:** 10.3390/ijms21061911

**Published:** 2020-03-11

**Authors:** Susanne Tonnemacher, Mikhail Eltsov, Burkhard Jakob

**Affiliations:** Department of Biophysics, GSI Helmholtzzentrum für Schwerionenforschung, 64291 Darmstadt, Germany; s.tonnemacher@gsi.de

**Keywords:** DNA repair, carbon ions, radiation-induced damage, chromatin structure, electron microscopy, CLEM, DNA and RNA cytochemistry, Osmium ammine B, ChromEMT, DNA-specific staining

## Abstract

Chromatin architecture plays major roles in gene regulation as well as in the repair of DNA damaged by endogenous or exogenous factors, such as after radiation. Opening up the chromatin might provide the necessary accessibility for the recruitment and binding of repair factors, thus facilitating timely and correct repair. The observed formation of ionizing radiation-induced foci (IRIF) of factors, such as 53BP1, upon induction of DNA double-strand breaks have been recently linked to local chromatin decompaction. Using correlative light and electron microscopy (CLEM) in combination with DNA-specific contrasting for transmission electron microscopy or tomography, we are able to show that at the ultrastructural level, these DNA damage domains reveal a chromatin compaction and organization not distinguishable from regular euchromatin upon irradiation with carbon or iron ions. Low Density Areas (LDAs) at sites of particle-induced DNA damage, as observed after unspecific uranyl acetate (UA)-staining, are thus unlikely to represent pure chromatin decompaction. RNA-specific terbium-citrate (Tb) staining suggests rather a reduced RNA density contributing to the LDA phenotype. Our observations are discussed in the view of liquid-like phase separation as one of the mechanisms of regulating DNA repair.

## 1. Introduction

Genomic DNA is packed in a highly regulated manner within an interphase nucleus to allow gene transcription or duplication of DNA during the S-Phase. To achieve the necessary compaction, the DNA double-helix is wrapped around octamers of core histones forming nucleosomes (reviewed in [1]). The interactions of neighboring nucleosomes, through exposed unfolded regions of core histones enriched in positively-charged residues together with linker histone binding, lead to further compaction (reviewed in [2]). This process is fine-tuned by covalent modifications of histones that encrypt epigenetic codes and recruit specific non-histone regulatory chromatin binders, such as polycomb group proteins (PcG) or heterochromatin protein 1 (HP1), which can result in dense chromatin states known as heterochromatin (reviewed in [3,4]). At the megabase pair scale, longer-distance chromatin interactions segment the interphase chromatids into domains, which can be visualized by imaging techniques and revealed by chromatin conformation capture methods as topologically associated domains (TADs), which eventually cluster, thus forming the interphase chromosome territory (reviewed in [5]). The chromatin domains occupy only a part of the nuclear volume, interspaced by other nuclear compartments, such as the nucleolus, nuclear speckles, and bodies, which are characterized by a low DNA density, whereas they are enriched in RNA and proteins (reviewed in [6]).

Although a detailed understanding of chromatin structure beyond the nucleosome level is still missing, the chromatin compaction plays an important role in the regulation of gene activity. Active genes are located in scattered, low-compacted domains, termed euchromatin (EC), whereas the inactive part of the genome is organized into compact heterochromatin (HC). Besides its role in gene transcription regulation, chromatin structure plays a major role in the repair of damaged DNA as well (reviewed in [7,8,9]). Efficient and correct DNA repair after, e.g., a radiation insult, is a prerequisite to maintaining genomic stability in mammalian cells [10]. However, especially the repair of double-strand breaks (DSBs) was shown to be heavily impacted by the chromatin environment (reviewed in [8,11]) and that, moreover, dramatic modifications arise in the chromatin surrounding a DSB especially in HC [12,13,14,15]. Both the induction of radiation induced DSBs [16,17] as well as the location of repair events and the utilization of repair factors was shown to be influenced by chromatin density, but also—depending on the type of damage induction or DSB density—by the cell cycle and utilized repair pathway [18]. Using various approaches, we have been able to show a localized depletion of heterochromatic DNA staining or changes in the fluorescence lifetime of DNA dyes in mouse chromocenters upon irradiation with charged-particles, indicating a local chromatin decompaction, which was accompanied by a rapid recruitment of DNA repair factors [13,19,20]. Recently, these results have been corroborated using transmission electron microscopy (TEM) showing electron translucent areas in the interphase nuclei of human fibroblasts at putative sites of ion traversals upon irradiation with carbon or calcium ions after embedding into LR White and uranyl acetate (UA) staining [21]. Interestingly, these electron translucent areas (subsequently termed LDA for Low Density Area) show a high density of DSB repair factors like 53BP1 [22] indicated by immunogold-labelling [21], thus confirming the presence of multiple radiation-induced DNA DSBs or clustered DNA lesions. Recent fluorescence microscopy studies on 53BP1 repair foci show properties of liquid-like phase separated domains [23], which could be an underlying mechanism for the observed changes in EM. Nevertheless, addressing the internal organization of these liquid-like phase separated domains and their relation within other nuclear components requires a higher resolution visualization.

The formation of LDAs revealed by the electron microscopy study of Timm et al. [21] matched in time with the formation of focal accumulation of repair factors visualized by immunofluorescence. Although a direct structural overlap between these two events was not provided, these data together with the results of immunogold labelling strongly indicated a distinct structural phenotype of DNA repair foci. It was hypothesized that the LDA pattern was a result of a local chromatin decompaction. Nevertheless, due to unspecific affinity of uranyl acetate to different cellular components, such as RNAs and proteins, this stain could not provide conclusive evidence for DNA reorganization linked to DNA repair.

In the present work, we combine correlative light and electron microscopy (CLEM), the localization of DNA repair foci with two DNA-specific EM staining techniques, ChomEMT, and Feulgen-type reaction with osmium ammine B (OA-B) to explore DNA arrangement therein. We show that the DNA/chromatin compaction state within DNA repair regions generally is lower than in HC, but comparable with euchromatic areas, thus DNA rearrangement cannot be solely responsible for the LDA pattern. We show that, contrary to the rest of nucleoplasm, LDAs have a low density of nuclear RNA-rich components, suggesting that low electron density of DNA repair regions results from the diminution of RNA-rich components rather than DNA/chromatin decompaction. On the basis of our observations, we hypothesize that DNA repair foci (IRIF) encapsulate damaged DNA through phase separation from nucleoplasm to provide favorable conditions for DNA damage repair.

## 2. Results

### 2.1. LDA Phenotype is Fixation and Embedding Independent

We applied CLEM to map DNA repair foci in resin embedded samples, using U2OS cells expressing 53BP1-GFP (Figure 1). These cells were irradiated with C or Fe ions and chemically fixed 1 h or 5 h after irradiation. Two fixation protocols were examined and compared: (i) 2% formaldehyde (PFA) and 0.05% glutaraldehyde (GA) in PBS^−/−^ to reproduce conditions of Timm et al. [21], and (ii) 2.5% glutaraldehyde in 0.1 M sodium cacodylate buffer (SCB), which is used in protocols for DNA-specific staining [24,25].

Fluorescence imaging of samples performed directly after the different types of fixation revealed the characteristic formation of radiation-induced foci (IRIF) indicated by the accumulation of 53BP1-GFP fluorescence at the DNA damaged domains after irradiation with C or Fe ions (Figure 1B,D, Figure 2B,D,F,H). The characteristics of the 53BP1 IRIF were in full accordance with live cell observations before fixation and with the ones obtained previously after particle irradiation [26,27]. The finding that this distinct fluorescence pattern, as well as sufficient signal intensity, was preserved after fixation indicated that both fixation protocols did not affect the 53BP1 distribution and the nuclear architecture at the detectable level. Fixed samples were imaged using confocal LM at low magnification (20× lens) (Figure 1A) to map cells in relation to the marker grid, for further re-localization at EM level; and at higher magnification, applying a high NA oil immersion lens (63×) (Figure 1B,D) to record three-dimensional maps of 53BP1-GFP foci inside the selected nuclei. After imaging, the samples were dehydrated with a series of acetone dilutions and embedded into Epon resin (see Materials and Methods) (Figure 1C).

Using this approach we could show that at the EM level, each 53BP1-GFP focus corresponded to a LDA in sections of resin-embedded samples after carbon ion irradiation stained with UA (Figure 1D/E). Importantly, very similar LDAs were found in samples after both fixation protocols and embedding in LR White as well as in Epon also after longer (5 h) post-irradiative incubation time and irradiation with heavier particles (Fe) (Figure 2). The internal content of LDA in all cases showed a fine fibrillar aspect without visible order. Such fixation and embedding robustness of structural features indicates that LDAs are not an artefact of a given EM protocol, but linked to long-lasting radiation-induced structural rearrangement in the nucleus.

The observed general internal structure of nuclei in EM sections stained with UA appeared to be quite similar after both protocols and is in agreement with previously published observations [21,28]. We could distinguish highly dense nucleoli and granules, whereas the overall density of the remaining nuclear content was relatively uniform, without clearly distinguishable HC domains (Figure 1E, Figure 2A,C,E,G). Interestingly, granules were mostly located inside or at the periphery of LDAs (Appendix A). The uniform staining density of the nuclear content visible after UA staining at EM level differs from the signal of the DNA-specific dye DRAQ5 [29] (chromatin) in the FM images. The later shows a complex organization of low-intensity areas (euchromatin) and heterochromatic domains well discernible around nucleoli, which is not visible in EM images stained with UA, thus confirming the unspecific staining of UA (Figure 2). Therefore, we have to conclude that UA staining cannot be used for the unambiguous assessment of DNA compaction.

In order to provide a control of the potential effects of over-expression of 53BP1 on LDAs, WT NIH/3T3 cells expressing only endogenous 53BP1 were irradiated with carbon ions using targeted irradiation with a cross pattern at the GSI microbeam setup [30]. LDA patterns similar to the ones described before for the U2OS cells were obtained in NIH/3T3 cells fixed with protocol (ii) at the sites of ion traversals (Appendix A). This observation, together with the previously published data [21], supports the general occurrence of LDAs in different cell types after irradiation with high linear energy transfer (LET) radiation.

### 2.2. DNA-Specific Stains Reveal EC-Like Areas within DNA-Repair Foci, but No LDA Pattern

In order to overcome the limitations of UA staining for judging on chromatin density, irradiated U2OS cells fixed with glutaraldehyde were further processed for ChromEMT, a recently developed method for chromatin-specific contrasting [24]. For this purpose, DNA was firstly stained with the DNA intercalating dye DRAQ5, then diaminobenzidine (DAB) polymerization was specifically induced around DNA molecules via photooxidation and DAB precipitations were stained by reduced osmium [24]. In contrast to UA staining, this DNA-specific stain revealed non-homogeneous distribution of DNA within the cell nuclei (Figure 3). It consists of dense heterochromatic patches located at the nuclear and nucleolar periphery, and in part within the nucleoplasm as well as lower density areas indicating euchromatic regions. The chromatin density correlates well with the DRAQ5 signal. ChromEMT staining revealed some background in the nucleoli and cell membranes, which is in agreement with published data [24] and results from some unspecific affinity of reduced osmium.

Surprisingly, irradiated U2OS cells show no LDA pattern after ChromEMT staining. Using the CLEM approach described before, we could precisely locate areas corresponding to the 53BP1 IRIF in serial thin sections, thus allowing a dedicated analysis of the irradiation sites (Figure 3, Appendix A). Sites of 53BP1 IRIF showed a relatively low density of DNA in comparison to HC patches, however their density was very similar to other euchromatic areas (Figure 3E,F).

To independently verify our results after ChromEMT staining, we applied DNA-specific osmium ammine B (OA-B). This stain is based on the Feulgen reaction, which is specific to DNA [31], thus avoiding unspecific contrasting of cellular components with the reduced osmium used in ChromEMT staining. The general contrast of chromatin after OA-B is lower compared to ChromEMT staining, because the reaction occurs only at the section surface. Nevertheless, the results of OA-B stain on irradiated U2OS cells were quite similar to ones obtained in ChromEMT stained samples: IRIFs showed a low DNA density, which was again comparable to non-irradiated euchromatin areas (EC-like (EC-l) structures, Figure 4).

Additionally, these observations could be reproduced again using WT NIH/3T3 cells (Appendix A), thus corroborating our findings. Importantly, these confirming results were obtained from the same samples which were used for UA staining showing a clear LDA pattern (Appendix A). This strongly supports our hypothesis that the observed LDAs are not purely DNA decondensation patterns.

### 2.3. Tomographic Analysis of Chromatin Organisation within DNA Repair Foci

To better understand the chromatin organization linked to DNA repair we performed dual-axis tomography on semi thin (2500 nm) sections of irradiated cells stained for ChromEMT. The OA-B method is not suitable for this task because of the location of the stain, preferentially at the section surface.

The areas for tomography data collection was targeted by confocal imaging using DRAQ5 DNA densities as fiducial markers (Appendix A). ChromEMT visualized a 3D-network of heterogeneous fibers and roundish dense bodies. The fibers appear very similar in irradiated areas and non-irradiated EC areas, whereas roundish domains prevailed in HC (Figure 5, Appendix A). The position and shape of 53BP1 IRIF were extracted from confocal fluorescence images, converted into two dimensional masks and overlapped with the slices of the EM-tomograms after adjustment of the magnification (Figure 5 white outline, Appendix A). Then, the tomogram was cut along the z-axis around the mask contour, the corresponding volume extracted, and the fiber thickness analyzed. To compare irradiated and non-irradiated EC, the IRIF contour masks (shape and area) were randomly placed over non-irradiated EC and volumes extracted, accordingly (Appendix A). For the HC analysis a similarly sized mask was used confining only HC-volumes (Appendix A).

To compare chromatin features quantitatively, we segmented the chromatin elements in the extracted volumes and measured their diameters using custom software developed in-house (see Materials and Methods, Appendix A). Diameter distributions obtained from three irradiated and non-irradiated EC areas from three individual cells were compared to each other and to HC areas from the same tomograms. The obtained fiber diameter distributions were nearly identical for irradiated areas and non-irradiated EC (Figure 6 green/red). Both showed a prevalence of 30–60 nm fibrillar elements and a median of ~46.5 nm for irradiated and of ~45.5 nm for non-irradiated EC. HC areas were enriched with elements of large diameter, thus increasing the median value to ~66.2 nm (Figure 6 purple).

### 2.4. Reduced RNA Density within DNA Repair Regions

Since chromatin within DNA repair foci was not different in density and organization from regular euchromatic regions, we decided to explore the contribution of another, non-chromatin nucleic acid constituent (RNA) to LDA formation. To map the distribution of RNA-rich nuclear components, we used terbium-citrate (Tb) staining introduced by the group of Fakan [32], who demonstrated selective affinity to RNA. Accordingly, the stain showed a high density accumulation within nucleoli (Figure 7A), whereas the areas corresponding to high DNA density at the border of nucleoli and close to the nuclear envelope showed lower density in comparison to UA staining (Appendix A). Importantly, the LDAs were present in terbium-stained sections and corresponded again to 53BP1 IRIF 1h after irradiation, as we could visualize using our CLEM approach (Figure 7A,B; see also Appendix A). RNA diminution at LDAs was also clearly visible at 5 h post-irradiation after LR White embedding (Appendix A). These results give a clear indication for the exclusion of RNA-rich components from the DNA repair regions. In agreement with that, we also noticed areas of a higher Tb contrast, which corresponded to low DNA density visualized by DRAQ5 staining, indicating again a selective binding to RNA (Appendix A). These results are corroborated by the staining of irradiated cells using the fluorescent RNA-specific dye Syto RNASelect Green, which supported the observation of a reduced density of RNA in 53BP1 IRIFs (Appendix A).

To assess this staining specificity in our preparations, we pre-treated sections with RNase prior to incubation with Tb. Even though these samples still show some staining density in nucleoli indicating an incomplete digestion of RNA in chemically fixed and resin embedded samples, LDAs completely vanished or appear strongly diminished after RNase treatment (Figure 7C&D). Together, these results provide clear evidence for the participation of RNA-rich components in UA stained samples and the contribution of RNA diminution in the formation of radiation induced LDAs.

## 3. Discussion

In this study, we explored the radiation-induced reorganization of cell nuclei after charged particle irradiation by applying CLEM, a combination of fluorescent light microscopy detection of IRIF at irradiation sites and electron microscopy techniques for selective staining of DNA and RNA. Our CLEM results clearly demonstrated that radiation-induced DNA repair foci, visualized by the fluorescence of 53BP1-GFP in U2OS cells, directly correspond to the LDAs visible in EM sections after non-selective UA staining (Figure 1 and Figure 2). We could confirm LDA formation in ion-irradiated WT NIH/3T3 cells, and thus demonstrate that these LDAs are not an artefact of overexpression of 53BP1-GFP. This is in agreement with the recent observation of LDAs in human dermal fibroblasts after irradiation with carbon or calcium ions which were interpreted as decondensed chromatin regions (DCRs) [21]. Although the LDA phenotype was initially reported after formaldehyde fixation and embedding into acrylic resin (LR White) [21], we show that the LDA phenotype also occurs after glutaraldehyde fixation and Epon embedding (Figure 2), the sample preparation protocol used for DNA selective staining ChromEMT and OA-B. Interestingly, the observed LDA phenotype after irradiation (1 to 5 h) is very similar to the one of persistent DNA damage foci (PDDF), which was described in mouse neurons after DSB generation [33]. PDDFs also accumulate 53BP1 and their structural appearance in TEM images is visually indistinguishable from our LDAs, although they were visualized after low temperature embedding into Lowicryl K4M, which is a much milder treatment than we used [33]. This observation further outlines the structural consistency and robustness of DSB DNA damage repair structures in relation to cell type and the sample preparation protocol.

To check the previous models of LDAs, linking their structural appearance to chromatin decondensation (termed as DCRs in [21]), we applied DNA-specific stains to assess the density and structural organization of chromatin within DNA repair foci. EM sample preparation protocols generally are based on chemical fixation and dehydration prone to aggregation of chromatin fibers, which hampers their applicability in interpretation of native chromatin organization, especially at the nucleosome fiber scale [34,35,36]. Nevertheless, bearing this limitation in mind, these methods still can be used for comparative analysis of chromatin organization in response of a treatment, in our case for radiation-induced DNA damage, in cells fixed with the same protocol. Importantly, a comparison of distributions and shapes of HC patches visible at the EM level in the 3D reconstruction of nuclear volume from serial thin sections (Appendix A), and confocal fluorescence images before DAB polymerization and embedding, did not reveal detectable chromatin rearrangements during ChromEMT procedure, thus underpinning the validity of our approach.

Our results from ChromEMT, while generally supporting open chromatin after high-LET irradiation, did not reveal any difference in density and chromatin organization between irradiated area and non-irradiated euchromatic areas of the cell (Figure 3). However, we cannot rule out that in addition to the absence of significant chromatin changes in damaged EC, heterochromatic areas are “euchromatized” and thus decompacted at damage sites. This would be in agreement with the observation in light microscopy where, on the basis of fluorescence intensity or fluorescence lifetime imaging (FLIM), only radiation-induced chromatin decompaction at heterochromatic sites after charged-particle traversals had been observed [13,20].

We acknowledge that the fiber thickness distribution assessed by ChromEMT in irradiated U2OS cells (median of around 45 nm) differs from previously reported values (mainly 12–24 nm fibers) for non-irradiated primary human small airway epithelial cells [24]. This disparity might be partially linked to cell type specific differences; however we attribute it to a stronger photooxidation and deposition of DAB in our experimental setup. The resulting stronger DAB deposition and accordingly larger diameters of chromatin fibers in our experiments did not hamper comparative analysis, and was also beneficial for an unambiguous discrimination of DNA structures from the general background of reduced osmium unavoidable in ChromEMT. Importantly, our ChromEMT results are corroborated by independent OA-B staining based on a completely different reaction chemistry [25,31,37] and pointing again to a euchromatin-like (EC-l) structure at the sites of particle traversals (Figure 4). To summarize, DNA-specific EM staining clearly showed that DNA decompaction cannot be solely responsible for the LDA phenotype.

Our results from terbium staining strongly suggest that LDAs mainly originate from a substantially lower density of RNA at IRIF. This observation and interpretation agree with a low density of RNA in PDDF, demonstrated by Mata-Garrido et al. [33], using a different cytochemical technique, EDTA regressive stain [38]. In this respect, our observations provide a new interpretation for the results of previous electron spectroscopy imaging (ESI) of 53BP1 IRIF [39,40] that revealed a lower phosphorus density and correspondingly, a lower concentration of nucleic acids therein in comparison to surrounding nuclear structures. Since, according to our results, the DNA density of DNA damage domains is very similar to the surrounding EC, the low phosphorus density observed in ESI must have resulted rather from RNA diminution. Supporting that view, immunofluorescence confocal microscopy showed no colocalization between IRIF and RNA-rich speckle domains or Cajal bodies, although they can be adjacent [41,42]. Even so, it has been shown that nascent transcript RNA might be utilized in error-free DSB repair at actively transcribed genes [43,44], the low RNA concentration within IRIF suggests a general sequestration of splicing and transcription components from these domains, thus leading to a local environment fully dedicated to DNA repair. In line with this view, DSB induction distal to the promoter of a reporter gene was shown to repress its transcription [45], and DSB induced an inhibition of RNA polymerase I [46], both in an ATM dependent manner. This would be beneficial for the protection of damaged DNA from further deterioration during transcription. Manfrini et al. [47] concluded that the inhibition of local transcription is a general response to DSB formation and is mediated by the resection of the break ends, which is a hallmark after high-LET radiation not only in G2 but also in G1-phase cells [48]. In this case it might prevent hybridization of putative exposed single stranded DNA with RNA, thus facilitating alternative end joining pathways. Interestingly, the initial formation of 53BP1 foci has been described to require the recruitment of RNA polymerase II and transcription of long non-coding DNA in the vicinity of a DSB [49,50,51] and processing them into small non-coding DNA-damage response RNA (DDRNA, reviewed in [52]). Although the evidence of these processes after high-LET irradiation have not yet been presented, such transcriptional activity is fully compatible with the open EC-l DNA organization within DNA damage domains. However, further research of the temporal aspects will be required to understand it in the context of the low RNA content, as observed in this study. We see several possible explanations: RNA synthesis occurs (a) at the beginning of IRIF formation and RNA is degraded or diffused away later or (b) at a much lower intensity in comparison to a transcriptional activity of the regular EC; (c) it is localized at the border of DNA damage domain or (d) besides the RNA specifically produced or utilized in the damage response, transcription is generally inhibited (at later times) in the larger domain of the IRIF [44], leading to a net loss of RNA. Although the detection of the transcriptional activity in these domains was out of the scope of the current study, results from Mata-Garrido et al. [33] indicated the absence of RNA polymerase II (phosphorylated on Ser 2, as a marker of the elongation) as well as transcription inside neuronal PDDF. Nevertheless, the transcription was shown to be intense at the border of PDDFs, thus supporting the third hypothesis (c).

Several recent studies revealed that IRIF show liquid-like properties of biomolecular condensates [13,23,50]. Formation of those condensates can be driven by liquid-liquid phase separation (LLPS) of the 53BP1 protein [23] and might enhance DSB end-joining probability by isolation of the break ends in confined space. The phase separation can further contribute to DNA damage repair by the modulation of macromolecular diffusion, thus favoring the accumulation of DNA repair factors inside IRIF and the exclusion of undesired components, such as transcription and splicing machineries. The observed roundish shape of LDAs, and the low RNA content of IRIF, are fully compatible with this hypothesis. Indeed, the strikingly “empty” appearance distinguishing IRIFs in EM images suggests a diffusion barrier, preventing the uncontrolled entrance of mobile macromolecules, which would otherwise equilibrate the IRIF density with the one of the crowded nucleoplasm. The nature of this barrier remains to be unraveled; however, our EM results suggest that it is unlikely to rely solely on the mere accumulation of 53PB1 or other proteins. The “empty” phenotype in TEM indicates the abundance of molecular components, which cannot be crosslinked by aldehydes during fixation and, thus, are extracted during dehydration and embedding steps of sample preparation for EM. Phosphoinositide lipids (PPIs) can be one candidate for such “extractable” components. Due to the absence of amino groups, they cannot be crosslinked by aldehydes, are enriched in the nucleus and, according to recent studies [53], accumulate quickly at DNA damage sites. Although RNA can in principle react with aldehydes, its crosslinking is very low in comparison to proteins [54], thus chemical fixation may have a low efficiency for the immobilization of small RNA species. Thus, the extraction of DDRNAs provides us with another hypothesis explaining LDA phenotype. In addition to extraction, conventional sample preparation for TEM might lead to the precipitation of poorly fixed components within IRIFs, we speculated that it might explain the abundance of the dense granules within them. Testing of this hypothesis and the further analysis of IRIF will require the application of cryo-electron microscopy methods to explore the IRIF structure in the absence of the putative extraction/precipitation artefacts.

Conclusion: Using CLEM and chromatin-specific contrasting, we demonstrated that the chromatin density and organization within ionizing radiation-induced DNA damage domains is very similar to EC, thus arguing against the previous models of the particular chromatin decondensation during DNA DSB repair, at least in euchromatic regions. The here described LDA phenotype of IRIFs is characterized by low RNA content and supporting models of LLPS in DNA repair domains after charged particle irradiation.

## 4. Materials and Methods

### 4.1. Cell Culture, Irradiation, and Fixation

Human osteosarcoma cells (U2OS) stably expressing 53BP1-GFP (kindly provided by C. Lukas, Copenhagen, Denmark) were grown on round photo-etched coverslips (28 mm in diameter, ibidi, Gräfelfing, Germany) in 35 mm petri dishes at 37 °C with 95% humidity and 5% CO_2_ in DMEM (Gibco, Carlsbad, CA, USA) + 10% FCS (Biochrom, Berlin, Germany) + 1 µg/mL Puromycin. The irradiation was done at the linear accelerator UNILAC of the GSI Helmholzzentrum für Schwerionenforschung GmbH with carbon (9.8 MeV/u, 170 keV/µm) or iron ions (7.0 MeV/u, 2875 keV/µm) at a fluence of 5×10^6^ particles/cm^2^. For the microbeam experiment, NIH/3T3 cells (ATCC, Manassas, VA, USA) were prepared and irradiated according to [30], using 4.8 MeV/u carbon ions (LET: 285 keV/µm). Fixation was done 1 h (carbon ions) or 5 h (iron ions) after irradiation, respectively. Two fixation methods were used: (i) 2% formaldehyde (PFA; Sigma-Aldrich, Saint-Louis, MO, USA) + 0.05% EM grade glutaraldehyde (GA; Science Services, München, Germany) in PBS^-/-^ (w/o calcium and magnesium; Biochrom, Berlin, Germany), 5min RT + 1h on ice; (ii) 2.5% GA + 5 mM CaCl_2_ in 0.1 M sodium cacodylate buffer (SCB; Serva, Heidelberg, Germany), 5 min RT + 1 h on ice, 3 × 5 min blocking with 10 mM glycine (Roth, Karlsruhe, Germany) in 0.1 M SCB.

### 4.2. Fluorescence Microscopy

Fluorescence microscopy was done after fixation, but before DAB polymerization, dehydration, and embedding. Mosaic screening of the GFP-signal of the whole coverslip was done automatically on a Nikon TiE spinning disc confocal microscope, using a 20× dry lens. Two channels (DRAQ5 and GFP) of the selected nuclei were recorded again at a higher resolution, using a 63x oil immersion lens, NA 1.3 at a Leica DMI 4000 B.

### 4.3. Electron Microscopy

#### 4.3.1. Embedding and Cutting for EM

After FM imaging with Leica DMI 4000 B, the samples were dehydrated and embedded into a resin. For LR White (Science Services, München, Germany) embedding, dehydration was done with ethanol (Roth, Karlsruhe, Germany) (5 min 30%, 10 min 50%, 15 min 70%, 20 min 90%, 2× 30 min 100%), followed by incubation with LR White/100% ethanol (1:1 mixture) overnight, 2x changes 100% LR White for 2 h. Then, the coverslips with cells were placed into a 35 mm Petri dish filled with fresh LR White, the dish cover was sealed by the application of a thin layer of LR White resin mixed with accelerator to reduce oxygen penetration inside the dish. After 30 min of the sealing resin layer polymerization at RT, the dishes were placed into an oven and cured at 55° for 48 h. After polymerization the coverslip was removed from the resin blocks through a careful cooling by contact with a metal rod that was pre chilled with liquid nitrogen.

For Epon embedding, dehydration was done with acetone (5 min 30%, 10 min 50%, 15 min 70%, 20 min 90%, 2× 30 min 100%; Electron Microscopy Sciences, Hatfield, PA, USA) and 2x 10 min with 100 % propylene oxide (Science Services, München, Germany). After dehydration, samples were incubated once in a 1:1 mixture of propylene oxide and Epon812 substitute (Sigma-Aldrich, Sent-Lois, MO, USA; 30 min) and twice with pure Epon812 substitute (2h and overnight). Polymerization of Epon was done at 65 °C for 48 h. Resin blocks were trimmed using diamond trimming knives (Diatome, Nidau, Switzerland). For TEM 100 nm sections and for TEM tomography, 250 nm sections were directly cut from the top side of the trimmed block with an ultra 35° diamond knife (Diatome, Nidau, Switzerland). Sections were collected on Formvar-coated (Electron Microscopy Sciences, Hatfield, PA, USA), palladium-copper slot grids (Science Services, München, Germany) and stained in 2% Uranyl acetate (Science Services, München, Germany) solution in methanol.

#### 4.3.2. ChromEMT Staining

The DNA-specific staining was done according to the original protocol published by Ou et al. [24]. Living cells were stained with 1 µM DRAQ5 (Thermo Fisher Scientific, Waltham, MA, USA) in OptiMEM (Gibco, Carlsbad, CA, USA) at 37 °C for 10 min, washed 2× 5 min with PBS^−/−^ and fixed. The coverslip with cells was transferred into a new plastic petri dish and bathed in 2.5 mM DAB (Sigma-Aldrich, Saint-Louis, MO, USA) in 0.1 M SCB. For the photooxidation of a larger area, DRAQ5 was excited at 615 nm provided by a CoolLED pE light source (BFI Optilas, Dietzenbach, Germany) set at 100% and applied via a light guide placed directly below the petri dish for up to 1 h at RT. A round area with a diameter of 5 mm was illuminated simultaneously. Cells were washed for 1 h (3× 20 min) in ice-cold 0.15 M SCB + 2 mM CaCl_2_ on ice and stained 1 h in 0.15 M SCB + 2 mM CaCl_2_ + 1.5% potassium ferrocyanide + 2% aqueous OsO_4_ (Science Services, München, Germany) on ice and subsequently washed 5× 3 min with ddH_2_O at RT, followed by dehydration and embedding in Epon, as described above (see 4.3.1).

#### 4.3.3. Staining with Osmium Ammine B (OA-B)

100 nm thick sections of Epon-embedded samples were collected on 200 mesh gold grids (Plano, Wetzlar, Germany). The grids were dried and incubated for 3 h at 70 °C to improve the section attachment, and cooled to RT. The grids were incubated with 5N HCl for 30 min at RT within 1.5 mL Eppendorf tubes, washed with ddH_2_O, air dried, and immersed into the osmium ammine B (Polysciences, Hirschberg, Germany) reagent activated exactly as described by [25]. After staining for 1 h at 37 °C, the grids were washed with ddH_2_O and air dried.

#### 4.3.4. RNA-Specific Staining with Terbium-Citrate (Tb)

The staining reagent was prepared as described by [32]. 100 nm thick sections of LR White embedded samples prepared according to protocol (i) were collected on Formvar-coated, palladium-copper slot grids. The grids were incubated on the reagent for 1 h. For control of the staining specificity, selected grids with sections were incubated on drops of 10 mg/mL RNase A (Sigma-Aldrich, Saint-Louis, MO, USA) in 50 mM Tris-HCl pH 7.0 at 37 °C for 24 h and washed with distilled water before Tb staining. The solution of RNase was boiled for 5 min to eliminate a risk of possible proteinase contamination and cooled down to 37 °C before use.

#### 4.3.5. Transmission EM

100 nm thick sections were collected on Formvar-coated slot grids and imaged in a transmission electron microscope (Jeol 2100Plus, 120 kV). For serial sections, Serial EM software (University of Colorado [55]) was used.

#### 4.3.6. EM Tomography

250 nm thick sections were collected on Formvar-coated slot grids for TEM tomography. Next, 15 nm colloidal gold particles (Sigma-Aldrich, Saint-Louis, MO, USA) were applied on the supporting film to serve as fiducial markers for tomogram alignment. The tomographic data collection was performed using Tecnai F30 transmission electron microscope (Thermo Fisher Scientific, Waltham, MA, USA) equipped with a US4000 CCD camera (Gatan, Pleasanton, CA, USA). Regions of interests (ROIs) were pre-exposed at 3000 e/Å^2^ to minimize the section drift and warping [56,57]. The dual-axis tomographic acquisition was done at 12,000× magnification (1.96 nm/pixel, camera binning 2) using Serial EM software (University of Colorado [55]) running a continuous tilt-scheme from −60° to +60° and a constant angular increment of 1.5°. Tilt series alignment and tomogram reconstruction were performed using ETOMO program of IMOD software package [58].

### 4.4. Data Analysis

The open source programs ImageJ (version 1.52a, NIH and LOCI, University of Wisconsin, USA) and IMOD (version 4.5.0, University of Colorado, USA) were used for general image analysis. Fiber thickness in the tomograms was determined using in-house written software, ImageD (David Eilenstein) (Appendix A). Briefly, image grey values were inverted and a 3D median filter applied to remove shot noise. A ROI was defined and selected. For chromatin fiber detection, a threshold was applied automatically inside the ROI and the resulting fibers binarized. Binarized fibers were skeletonized, center-points determined, and the diameter of the fibers analyzed on the basis of the shortest distance between center-points and boundary.

## Figures and Tables

**Figure 1 ijms-21-01911-f001:**
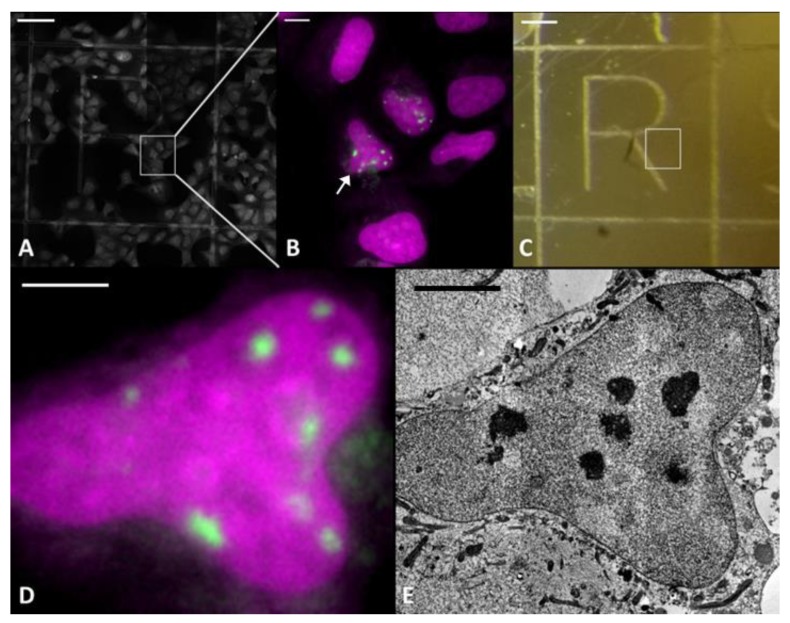
Defining region of interests (ROIs) and relocalization (CLEM): Fluorescence mosaic imaging of the whole grid was done automatically with a 20× dry lens (**A**). ROIs were defined by 53BP1-GFP expressing cells showing radiation induced repair foci (**A**,**B** and **D**) and imaged again at higher resolution using a 63x oil immersion objective (**B**). After the cells were embedded and removed from the glass coverslip, the resin block shows the imprint of the grid (**C**) and the ROI can be relocated (white box). Next, 100 nm sections were stained with UA and by the aid of the imprinted grid cells were relocated in the cut sections allowing imaging of the same nucleus in EM. FM (**B** arrow, **D**) and TEM (**E**). Scale bar: **A**&**C** 100 µm; **B** 10 µm; **D**&**E** 5 µm.

**Figure 2 ijms-21-01911-f002:**
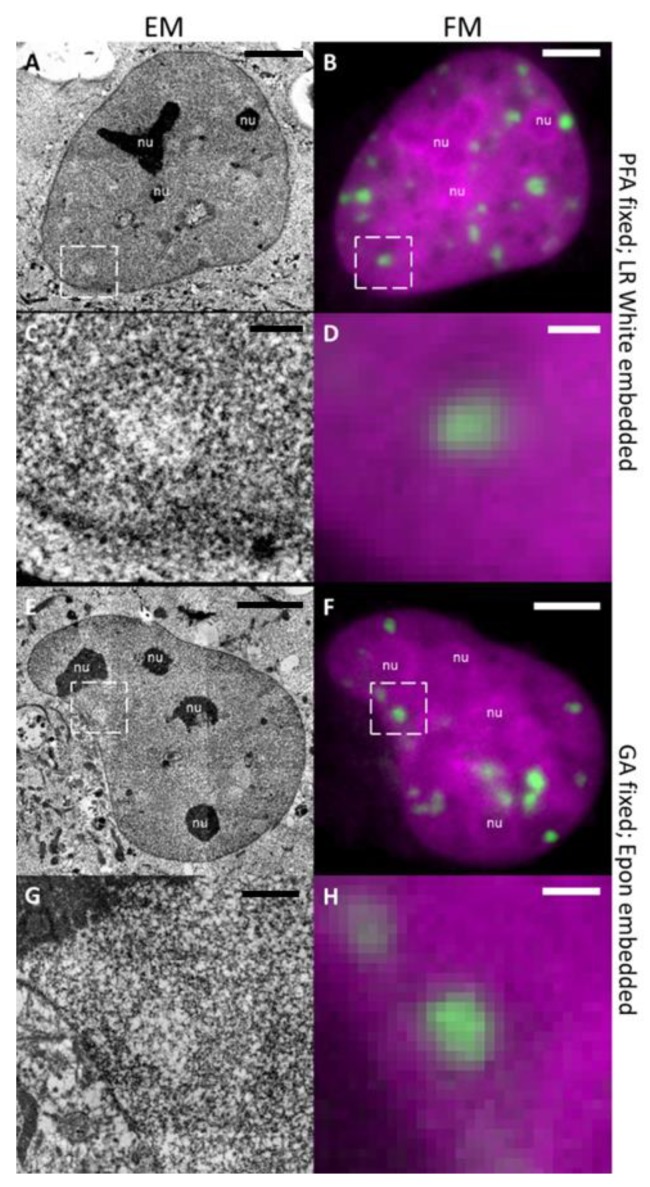
LDA phenotype is tolerant to different fixation and embedding conditions. Samples were PFA fixed and embedded in LR White (**A**–**D**) or GA fixed and embedded in Epon (**E**–**H**). Both samples clearly show LDAs in TEM images (left column). The whole nucleus imaged with TEM (**A**,**E**) or with FM (**B**,**F**) is shown. In (**C**,**G**) and (**D**,**H**) a single LDA (dashed white box of (**A**,**E**) and (**B**,**F**)) is magnified. All LDAs in the TEM images correlate with 53BP1-GFP IRIFs (green) of the FM images. Scale bar: (**A**,**B**,**E**&**F**) 5 µm, (**C**,**D**,**G**&**H**) 1 µm.

**Figure 3 ijms-21-01911-f003:**
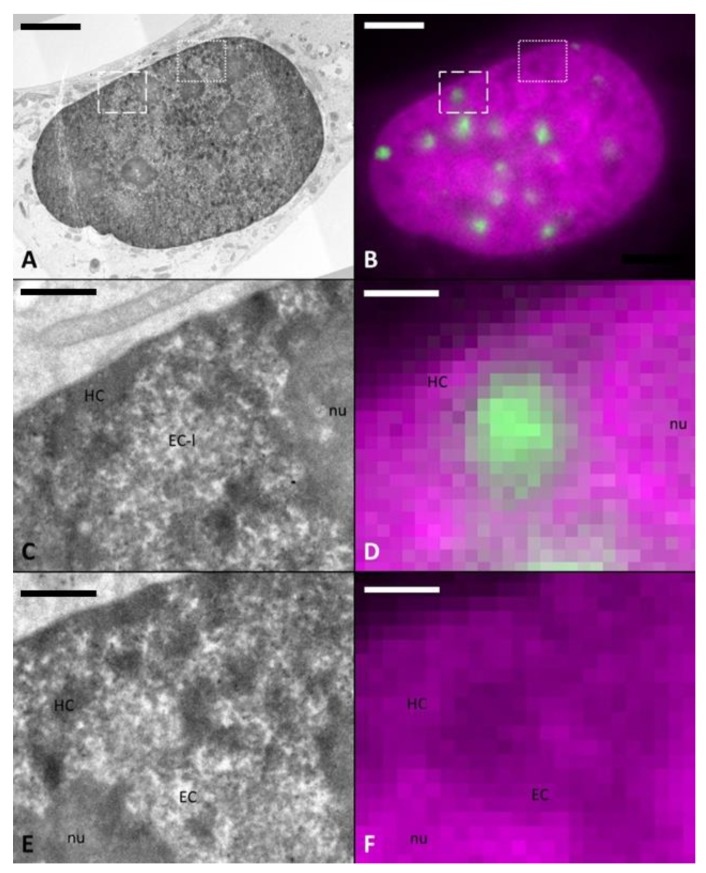
Absence of LDAs at damage site after chromatin-specific contrasting (ChromEMT). TEM image of a DNA-specific (ChromEMT) stained sample (**A**) correlated with FM image (**B**) with DNA (magenta) and 53BP1 (green). With ChromEMT staining, no LDA phenotype is visible. However, the irradiated area (dashed box) shows a structure (**C** with correlated FM image (**D**)) similar to euchromatin (EC) (doted box) of a non-irradiated area (**E**, correlated FM image (**F**)). Heterochromatin (HC) is clearly visible as more dark and dense structures mainly along the border of the nucleus and the nucleoli (nu) (**A**,**C**,**E**). Scale bar: (**A**,**B**) 5 µm, (**C**–**F**) 1 µm.

**Figure 4 ijms-21-01911-f004:**
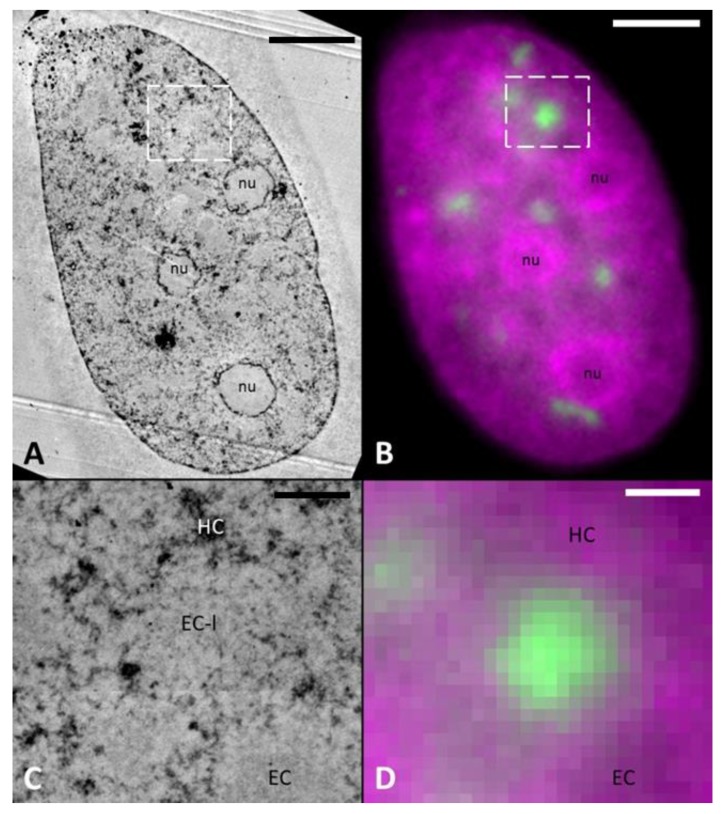
DNA-specific contrasting with osmium ammine B (OA-B). TEM image of OA-B stained sample (**A**). (**B**) Shows the same cell in fluorescence microscopy (53BP1 (green) and DNA (magenta)). Similar to ChromEMT stained sample (Figure 3) no LDA phenotype is visible. 53BP1 focus (**D**, green) correlates to a EC-like (EC-l) structure (**C**) and shows a similar DNA density to regular EC. Scale bar: (**A**,**B**) 5 µm, (**C**,**D**) 1 µm.

**Figure 5 ijms-21-01911-f005:**
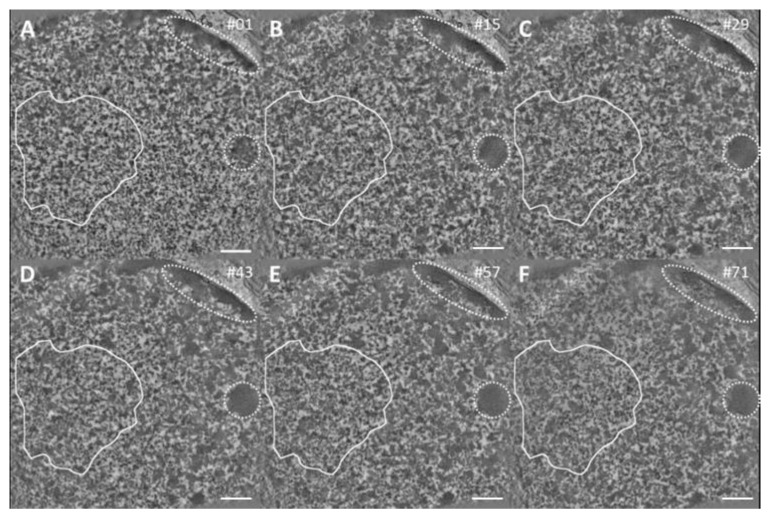
Tomography of an irradiated area stained by ChromEMT. A gallery of 2 nm thick tomographic slices extracted from a dual axis tomographic reconstruction of 250 thick section of an irradiated U2OS cell. The chromatin structure shows no obvious difference between the inside and outside of the damage site (white outline) through the whole volume. Roundish domains (inside doted areas) indicating HC. For further comparison see Appendix A which shows the tomography of a non-irradiated area. Slice numbers shown in the upper right corner. All scale bars: 500 nm. Slices thickness **2** nm.

**Figure 6 ijms-21-01911-f006:**
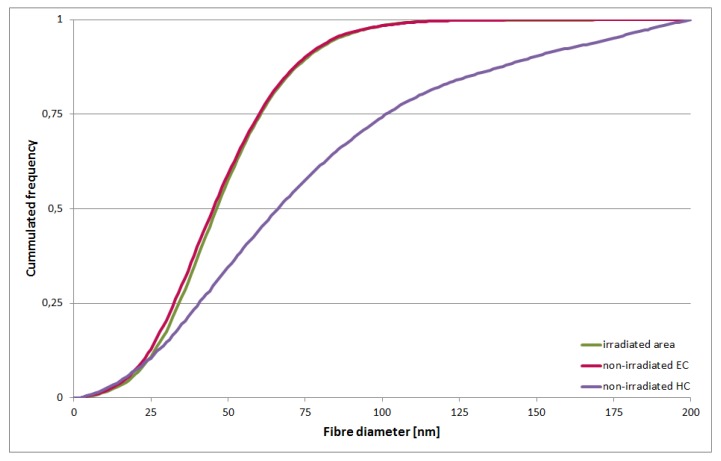
Cumulative histogram of fiber diameter of irradiated areas (green), non-irradiated EC (red), and HC (purple): Irradiated area shows a distribution of fiber diameters very similar to non-irradiated EC. The HC curve clearly deviates; it shows a higher proportion of fibers with a larger diameter. Each curve contains pooled data and was normalized to the volume. *N* = 3 images.

**Figure 7 ijms-21-01911-f007:**
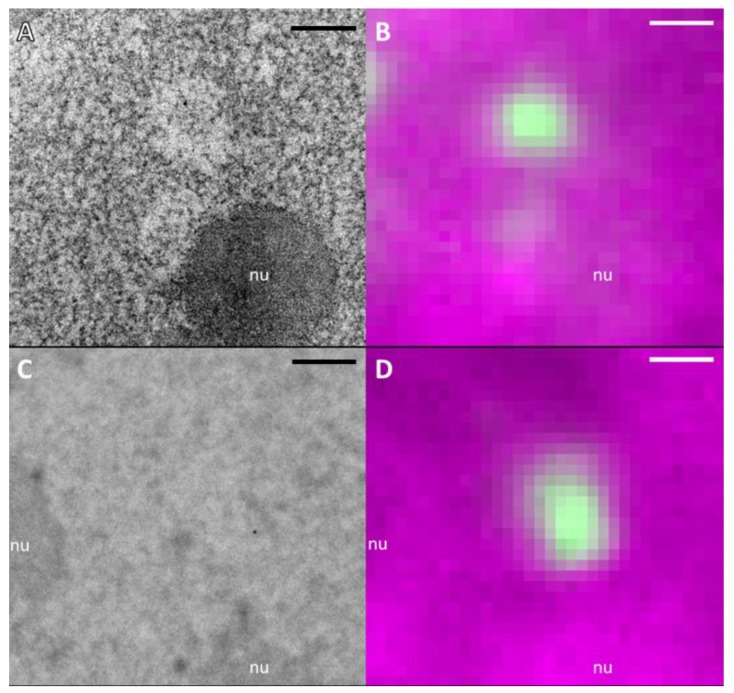
LDA phenotype at damage sites of RNA-specific stained (terbium-citrate; Tb) sections vanish after RNase treatment. Sections of U2OS nuclei irradiated with carbon ions, fixed 1 h after irradiation and stained with Tb (**A**) show LDAs correlating with 53BP1 signal (**B**, green). After 24 h treatment with RNase no LDAs are visible (**C**) at the damage site (**D**). All scale bars: 1 µm.

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
