# Peer review of "Correlative Light and Electron Microscopy (CLEM) Analysis of Nuclear Reorganization Induced by Clustered DNA Damage Upon Charged Particle Irradiation"

_ijms, 2020, doi:10.3390/ijms21061911_

Round 1

Reviewer 1 Report

Line 23: Define UA in UA staining

Line 33: Remove “e.g.” or revise the sentence

Line 42: …..chromatin conformation capture methods “such” as…..

Line 54: Period missing here and at multiple other lines. Carefully revise.

Line 145: What is DRAQ5? Describe here.

Figure 7: At what time point after radiation treatment the cells were fixed?  Provide kinetics of RNA exclusion after radiation treatment. Also use RNA specific Provide more images and statistical analysis for RNA exclusion at DNA repair foci. I would suggest to show comparative images with terbium staining and RNA specific fluorescent probe to confirm the LDAs have RNA exclusion phenotype.

Multiple current studies (such as Michelini F et al, Nat Cell Biol. 2017) have suggested that non-coding RNAs are required for 53BP1 foci formation. It is not clear how this author’s findings refute the published evidence of role of RNA in DNA damage response.

Author Response

We thank the reviewer for his constructive suggestions and comments. We carefully rephrased our statements regarding the RNA intensity at sites of ion traversals. Especially larger parts of the Discussion were rewritten including additional references to clarify the misconception of a contradiction between the observed LDA phenotype in EM and the published findings of utilisation of in the DNA damage response. Additionally, we provide several hypothesis for explanation. To support our EM data, we now added a new supplemental figure (S11) showing a general net loss of RNA at sites of ion traversals 4h after irradiation using fluorescence microscopy and a RNA specific stain. Also we added more Terbium-CLEM data (Supplemental Figures S9 and S10 + arrows in S8) corroborating our observation previously shown in Fig. 7. Below you can find a point by point response to the reviewer’s comments:

Line 23: Define UA in UA staining

This is now done in line 23 (in version without mark-ups)

Line 33: Remove “e.g.” or revise the sentence

Done as suggested

Line 42: …..chromatin conformation capture methods “such” as…..

“such” was deleted

Line 54: Period missing here and at multiple other lines. Carefully revise.

We apologise for that typos. We carefully checked and added the missing or removed the surplus full stops.

Line 145: What is DRAQ5? Describe here.

DRAQ5 is a cell-permeable DNA-specific deep red fluorescent dye (deep red‐fluorescing bisalkyl-aminoanthraquinone number five). We now added the missing information in the results section (line 144) which reads now “ …the signal of the DNA-specific dye DRAQ5 [29] (chromatin) in the FM images” and added a new reference (now [29] Smith, P.J., Blunt, N., Wiltshire, M., Hoy, T., Teesdale‐Spittle, P., Craven, M.R., Watson, J.V., Amos, W.B., Errington, R.J. and Patterson, L.H. (2000), Characteristics of a novel deep red/infrared fluorescent cell‐permeant DNA probe, DRAQ5, in intact human cells analyzed by flow cytometry, confocal and multiphoton microscopy. Cytometry, 40: 280-291.

Figure 7: At what time point after radiation treatment the cells were fixed? Provide kinetics of RNA exclusion after radiation treatment. Also use RNA specific Provide more images and statistical analysis for RNA exclusion at DNA repair foci. I would suggest to show comparative images with terbium staining and RNA specific fluorescent probe to confirm the LDAs have RNA exclusion phenotype.

Multiple current studies (such as Michelini F et al, Nat Cell Biol. 2017) have suggested that non-coding RNAs are required for 53BP1 foci formation. It is not clear how this author’s findings refute the published evidence of role of RNA in DNA damage response.

We are thankful to the reviewer for this comment. Our study does not question the general role of RNA for the damage response, it is simply out of scope of the paper. Our goal is to understand the IRIF structure. Indeed, there is no contradiction between our observations and published observations of transcription at the vicinity of DSB and synthesis of non-coding DNA. Our results do not exclude the synthesis and the presence of specific non-coding RNAs within IRIF, they just indicate that the general concentration of RNA therein is lower after chemical fixation 1-5h after irradiation and we demonstrated it with RNA-specific staining techniques, previously with Terbium chloride, and now also got supporting results from RNA-specific fluorescent dye Syto RNASelect Green, which is now given as Supplemental Figure S9. and stated in the Results line 256ff: “These results are corroborated by staining of irradiated cells using the fluorescent RNA-specific dye Syto RNASelect Green, which supported the observation of a reduced density of RNA in 53BP1 IRIFs (Supplementary Figure S11).

We also added the missing time point (1h) into the Results (line 250) and figure legend of Fig. 7 (“Sections of U2OS nuclei irradiated with carbon ions, fixed 1h after irradiation and stained with Terbium-Citrate (A) show LDAs correlating with 53BP1 signal (B, green). “.

Additionally to Figure 7 showing the reduced RNA at damage sites 1h after irradiation, we now provide supplementary RNA specific Terbium stained EM images showing more examples of the diminished contrasting after Epon embedding (Supplemental Figure S9). Also we indicated the lower RNA staining at sites of 53BP1 IRIF in Supplemental Figure S8 showing an different nucleus (1h Epon). In addition, we added now a different (%h) time-point as Supplemental Figure S10 showing Terbium staining after LR White embedding, again clearly indicating lower RNA content at damage sites. These additional Figures are now addressed in the Results (line 249ff): “…Importantly, the LDAs were present in terbium stained sections and corresponded again to 53BP1 IRIF 1h after irradiation, as we could visualise using our CLEM approach (Figure 7 A/B ; see also Supplemental Figures S8 and S9). RNA diminution at LDAs was also clearly visible at 5h post-irradiation after LR White embedding (Supplemental Figure S10). These results gives a clear indication….”

We agree that providing kinetics of RNA concentration at damage sites upon irradiation at high resolution in EM would be intriguing, but is beyond the scope of this manuscript and cannot be performed in adequate time frame. However, we might pick up this stimulating suggestion for future studies.

We can only speculate about the reasons why the RNA concentration is lower within LDA and we provided three hypotheses in the Discussion aiming to explain the diminution of RNA in the light of the published data supporting the role of transcription of non-coding RNA for IRIF formation. In this context we have rewritten the paragraph in the Discussion and implemented two additional new references as suggested (now [51] Michelini et al. Damage-induced lncRNAs control the DNA damage response through interaction with DDRNAs at individual double-strand breaks. Nat. Cell Biol. 2017 and [52] Bader et al. The roles of RNA in DNA double-strand break repair. Br. J. Cancer 2020).

We changed the section in the discussion which reads now (line 344ff): “….Interestingly, initial formation of 53BP1 foci has been described to require recruitment of RNA polymerase II and transcription of long non-coding DNA in the vicinity of a DSB [49,50,51], and processing them into small non-coding DNA-damage response RNA (DDRNA, reviewed in [52]). Although the evidence of these processes after high-LET irradiation have not yet presented, such transcriptional activity is fully compatible with the open euchromatin-like DNA organisation within DNA damage domains. However further research of the temporal aspects will be required to understand it in context of the low RNA content as observed in this study. We see several possible explanations: RNA synthesis occurs (a) at the beginning of IRIF formation and RNA is degraded or diffused away later or (b) at a much lower intensity in comparison to a transcriptional activity of the regular euchromatin; (c) it is localised at the border of DNA damage domain near the putative LLPS boundary or (d) besides the RNA specifically produced or utilised in the damage response, transcription is generally inhibited (at later times) in the larger domain of the IRIF [44], leading to a net loss of RNA. Although the detection of the transcriptional activity in these domains was out of the scope of the current study….

Unfortunately, the discussion on this point cannot go deeper, because it would require data on the density (concentration) of DSB-induced non-coding RNA versus the general nucleoplasmic one. To our knowledge, such data are not available. Furthermore, no specific evidence for the DSB-induced RNA transcription and the presence of specific non-coding RNAs in IRIFs after high-LET irradiation are present to date. High-LET is an exceptionally harsh DNA damage with a limited repair potential. In this context, our case is closer to persistent DNA damage foci in neurons, for those no transcriptional activity was revealed.

To completely exclude chances for misinterpretation, we added into the revised manuscript an additional hypothesis (line 378ff), which addresses the extraction of DDRNAs from IRIF during sample preparation, because these small molecules have chances to escape from immobilisation by chemical fixation: ”Although RNA can in principle react with aldehydes, its crosslinking efficiency is very low in comparison to proteins [54], thus chemical fixation may not be sufficient for the immobilization of small RNA species, thus extraction of DDRNAs provides us with another hypothesis explaining LDA phenotype.

We accept that some wording describing our results might be understood as a complete exclusion of the transcription machinery and RNA from IRIFs, and this could be misleading for the reader. To avoid misunderstanding, we revised the corresponding sentences:

Line 24f changed to “RNA specific terbium citrate staining suggests rather a reduced RNA density contributing to the LDA phenotype

Line 87f changed to “LDAs have low density of nuclear RNA-rich components, suggesting that low electron density of DNA repair regions results from diminution of RNA-rich components rather than DNA/chromatin decompaction.”

Line 241 changed to “2.4. Reduced RNA density within DNA repair regions

Line 323 changed to “Our results from terbium staining strongly suggest that LDAs mainly originate from a substantially lower density of RNA at IRIF.

Line 331 “RNA exclusion” was replaced by “RNA diminution”.

Line 389f (Conclusion): “The here described LDA phenotype of IRIFs is characterised by low RNA content and supporting models of LLPS in DNA repair domains after charged particle irradiation.

Reviewer 2 Report

Authors investigated electron microscopy and fluorescent microscopy to analyze clustered DNA damages induced by heavy ions and the association of euchromatin and heterochromatin. 

I don't have major concerns for this manuscript.

I have one minor point.

PBS-/-, I guess authors want to say PBS minus?

Author Response

We thank the reviewer for his positive evaluation of our manuscript.

I don't have major concerns for this manuscript.

I have one minor point.

PBS-/-, I guess authors want to say PBS minus?

PBS-/- indicates the absence of both Calcium and Magnesium.

this is now indicated in line 403 as well as in the “Abbreviations”

Round 2

Reviewer 1 Report

The authors' satisfactorily responded to all the review comments.